# Impact of Metastatic Pattern on Survival in Patients with Posterior Uveal Melanoma: A Retrospective Cohort Study

**DOI:** 10.3390/cancers16193346

**Published:** 2024-09-30

**Authors:** Tine G. Hindso, Peter S. Jensen, Mette B. Sjøl, Kristoffer Nissen, Camilla W. Bjerrum, Eric von Benzon, Carsten Faber, Steen F. Urbak, Marco Donia, Inge M. Svane, Eva Ellebaek, Steffen Heegaard, Karine Madsen, Jens F. Kiilgaard

**Affiliations:** 1Department of Ophthalmology, Copenhagen University Hospital—Rigshospitalet, Blegdamsvej 9, 2100 Copenhagen Ø, Denmark; mette.marie.bagger.sjoel@regionh.dk (M.B.S.); kristoffer.nissen@regionh.dk (K.N.); carsten.faber@regionh.dk (C.F.); steffen.heegaard@regionh.dk (S.H.); jens.folke.kiilgaard@regionh.dk (J.F.K.); 2Department of Ophthalmology, Aarhus University Hospital, Palle Juul-Jensens Boulevard 99, 8200 Aarhus N, Denmark; petjee@rm.dk (P.S.J.); steeurba@rm.dk (S.F.U.); 3Department of Radiology, Copenhagen University Hospital—Rigshospitalet, Blegdamsvej 9, 2100 Copenhagen Ø, Denmark; camilla.wium.bjerrum.01@regionh.dk; 4Department of Clinical Physiology and Nuclear Medicine, Copenhagen University Hospital—Rigshospitalet, Blegdamsvej 9, 2100 Copenhagen Ø, Denmarkkarine.madsen@regionh.dk (K.M.); 5National Center for Cancer Immune Therapy (CCIT-DK), Department of Oncology, Copenhagen University Hospital—Herlev and Gentofte, Borgmester Ib Juuls Vej 13, 2730 Herlev, Denmark; marco.donia@regionh.dk (M.D.); inge.marie.svane@regionh.dk (I.M.S.); eva.ellebaek.steensgaard@regionh.dk (E.E.); 6Department of Pathology, Copenhagen University Hospital—Rigshospitalet, Blegdamsvej 9, 2100 Copenhagen Ø, Denmark

**Keywords:** uveal melanoma, choroidal melanoma, ciliary body melanoma, ocular melanoma, eye cancer, metastasis, survival

## Abstract

**Simple Summary:**

In this study, we investigated whether the anatomical location of metastases from posterior uveal melanoma affects survival. We found that patients with newly diagnosed metastatic posterior uveal melanoma who only have extrahepatic metastases had a significantly longer survival compared to patients with liver metastases. This insight could help clinicians improve the prediction of patient outcomes and enhance the selection of patients in clinical trials.

**Abstract:**

**Background/Objectives**: Metastatic posterior uveal melanoma (PUM) is one of the deadliest types of melanomas. Though the median survival is short, some patients with metastatic disease live for a long time. In this study, we investigated whether the anatomical location of the metastatic lesions is associated with differences in survival. **Methods**: One hundred and seventy-eight patients with metastatic PUM with baseline whole-body imaging were retrospectively included. The patients were divided into three groups based on the anatomical location of metastases: (1) exclusive liver metastases (hepatic pattern), (2) both hepatic and extrahepatic metastatic lesions (hepatic–extrahepatic pattern), and (3) exclusive extrahepatic lesions (extrahepatic pattern). Survival was investigated using Kaplan–Meier plots, log-rank test, and the Cox proportional hazard model. **Results**: In total, 95 patients (53%) presented with hepatic pattern, 66 patients (37%) presented with hepatic–extrahepatic pattern, and 17 patients (10%) presented with extrahepatic pattern. Overall survival was significantly longer in patients with extrahepatic pattern (median 17.0 months) compared to those with hepatic pattern (median 11.0 months) and hepatic–extrahepatic pattern (median 7.0 months) (*p* < 0.001, log-rank test). Multivariate Cox regression analysis showed increased hazard ratios (HR) for hepatic pattern (HR 2.37, 95% CI 1.08–5.17, *p* = 0.031) and hepatic–extrahepatic pattern (3.25, 95% CI 1.42–7.41, *p* = 0.005) compared to extrahepatic pattern. Most patients with hepatic (95%) and hepatic–extrahepatic patterns (82%) were diagnosed with metastases by liver ultrasonography screening, whereas 81% of patients with extrahepatic pattern developed symptoms that led to the diagnosis. **Conclusions**: Extrahepatic pattern was associated with prolonged survival in patients with metastatic PUM, despite there being a larger proportion of symptomatic patients. It is therefore important to consider the anatomical location of the metastatic lesions when stratifying patients into clinical trials.

## 1. Introduction

Posterior uveal melanoma (PUM) is a rare melanoma subtype that arises in the ciliary body and the choroid. It is the most common primary intraocular malignancy, with the highest incidence rates, above 8.0 cases per million person years occurring in Northern European countries and in Australia and New Zealand [1,2]. PUM has a high tendency to metastasize, primarily to the liver [3,4], and about half of the patients die due to metastatic disease [5]. The size of the eye tumor, extraocular growth, ciliary body involvement, and chromosomal and genetic alterations in the primary tumor, such as monosomy 3 and mutation in *BAP1*, increase the risk of metastasis [6,7,8,9]. After the development of metastatic disease, the median survival is about 12 months [10]. Though survival is short, some patients with metastatic PUM live for several years [11], indicating that there might be patient subgroups with a less aggressive phenotype. Over the last few decades, there has been an increased focus on personalized risk stratification and identification of prognostic factors among patients with metastatic PUM [12,13,14,15,16]. A better risk stratification of the metastatic patients could facilitate relevant enrollment of patients in randomized clinical trials and potentially improve prognostication for the individual patient. Several biomarkers predicting survival from the onset of metastatic disease have been proposed [13,14,15,16]. Increasing size of the largest diameter of the largest metastatic lesion (LDLM) is incorporated as a negative prognostic factor for survival in the 8th Edition of The American Joint Committee on Cancer (AJCC) staging system (M1a: LDLM ≤ 3.0 cm; M1b: LDLM 3.1–8.0 cm; M1c: LDLM ≥ 8.1 cm) [12]. Performance status, liver enzyme levels, number of liver metastases, and disease-free interval have also been associated with survival in metastatic PUM patients [13,14,15,16].

It is well known that metastases from PUM disseminate to the liver in most cases, whereas only a minority of patients present exclusively with extrahepatic metastases [3,4]. Emerging evidence indicates that the metastatic pattern at diagnosis might be associated with prognosis [3,4,11,17,18,19,20,21,22]. Several small studies [3,11,17,18,19,20,21,22] and one recent larger study [4] have suggested that exclusive extrahepatic metastases are associated with prolonged survival. In these studies, data about the anatomical location of metastases were collected from medical records, liver ultrasonography, liver magnetic resonance imaging (MRI), abdominal computed tomography (CT), and, only to a limited extent, whole-body imaging (CT or positron emission-computed tomography (PET/CT)). Hence, extrahepatic involvement might be underrepresented in these cohorts. We therefore set out to investigate whether survival depends on the anatomical location of the metastatic lesions using whole-body imaging performed at the time of metastatic diagnosis.

## 2. Materials and Methods

### 2.1. Patients

Uveal melanoma patients treated at The Department of Ophthalmology, Copenhagen University Hospital, Copenhagen, Denmark are registered in the Copenhagen Epidemiological Uveal Melanoma Study (COEUS) database [23]. Based on chromosomal risk profiling, high-risk primary uveal melanoma patients are offered screening with regular liver ultrasonography every 6 months for 5 years, and then once yearly for a further 5 years. Patients that develop metastatic PUM are offered imaging work-up to assess the extent of dissemination, with CT or PET/CT of at least the chest and abdomen. Data on all patients diagnosed with primary PUM at Copenhagen University Hospital, Copenhagen, Denmark between 2000 and 2020 were identified from the COEUS database and the respective medical records were retrieved. Development of metastatic disease was last checked on 31 December 2022 (corresponding to at least two years of follow-up from primary diagnosis). All patients who had an MRI, CT, and/or PET/CT of at least both the chest and the abdomen performed within five weeks from diagnosis of metastatic PUM were retrospectively included. Patients without available imaging or with imaging performed later than five weeks from metastatic diagnosis were excluded.

### 2.2. Clinical and Imaging Data

Images and imaging reports were assessed via the local repository at Copenhagen University Hospital, Copenhagen, Denmark the local repository at Aarhus University Hospital, Aarhus, Denmark or retrieved from the local departments of clinical physiology and nuclear medicine or radiology departments throughout Denmark. Data on the anatomical location of metastases and the LDLM were collected from imaging reports (MRI, and/or CT, and/or PET/CT). When imaging reports were not available or the LDLM was not measured, two of the authors (EB and CWB) re-examined the scans. Extrahepatic metastases verified by other diagnostic methods than imaging (e.g., a skin biopsy performed within five weeks after metastatic diagnosis) were also registered. Oncological treatment data were retrieved from the Danish Metastatic Melanoma Database (DAMMED), a research database including all Danish metastatic melanoma patients [24]. First-line treatment with the combination of ipilimumab and nivolumab (ipi+nivo) was included as a treatment variable. The patients were staged according to the AJCC 8th Edition cancer staging system for patients with metastatic PUM [12]. The vital status was last checked on 31 December 2023 (corresponding to a minimum of 12 months of follow-up from metastatic diagnosis). The cause of death was evaluated using clinical charts and data from the Danish Register of Causes of Death [25]. Approval from the Regional Ethics Committee for the Capitol Region of Denmark was obtained with dispensation for informed consent (protocol number: H-21015415, 8 July 2021), and the study adhered to the tenets of the Declaration of Helsinki [26].

### 2.3. Statistics

The endpoint was overall survival, defined as the time from the date of the first metastatic lesion diagnosed on imaging to the date of death or last follow-up. None of the patients included were lost to follow-up. The disease-free interval was defined as the time from primary diagnosis until the date of the first metastatic lesion diagnosed on imaging. Survival curves were visualized using Kaplan–Meier plots and compared with log-rank tests. The Cox proportional hazard model was used to estimate hazard ratios (HR). Variables included in the multivariate Cox regression were chosen based on clinical relevance. The assumption of proportional hazards was evaluated for all variables using the cumulative score process test and cumulative martingale residuals. Non-proportional variables were included as strata in the multivariate Cox model. The Kruskal–Wallis test was used to check for statistical differences among multiple groups. Statistical analysis was conducted using RStudio (version 2023.06.0) (Rstudio Team, 2023) and the packages: “survival” (version 3.5.5), “survminer” (version 0.4.9), “mets” (version 1.3.2), “timereg” (version 2.0.5), “stats” (version 4.3.0), “ggplot2” (version 3.4.4), and “dplyr” (version 1.1.2). A *p*-value below 0.05 was considered significant.

## 3. Results

### 3.1. Patient Characteristics

In total, 252 out of 795 patients (32%) with PUM developed metastatic disease during follow-up. Seventy-one patients (28%) with metastatic disease were excluded due to unavailability of relevant imaging. Further, three patients (2%) were excluded due to the simultaneous occurrence of histopathologically verified metastases from other primary cancers. In total, 178 patients were included in this study (Appendix A, Table 1). A total of 95 patients (53%) had metastatic disease exclusively to the liver (hepatic pattern), 66 patients (37%) had both hepatic and extrahepatic metastases (hepatic–extrahepatic pattern), and 17 patients (10%) had exclusive extrahepatic metastases (extrahepatic pattern).

The median disease-free interval was 29.0 months for patients with hepatic pattern, 25.0 months for patients with hepatic–extrahepatic pattern, and 62.0 months for patients with extrahepatic pattern (Table 1). The diagnosis of PUM metastases was histopathologically verified in all 95 patients (100%) with hepatic pattern, in 61 patients with hepatic–extrahepatic pattern (92%), and in all 17 patients (100%) with extrahepatic pattern. Metastases were present at the time of primary diagnosis (AJCC stage IV) in 4 out of 95 patients (4%) with hepatic pattern, in 2 out of 66 patients (3%) with hepatic–extrahepatic pattern, and in 1 out of 17 patients (6%) with extrahepatic pattern.

### 3.2. Survival Analysis

Metastatic uveal melanoma was the only cause of death. Patients with extrahepatic pattern had a significantly longer overall survival (median 17.0 months) compared to patients with hepatic pattern (median 11.0 months) and to patients with hepatic–extrahepatic pattern (median 7.0 months) (Table 1, Figure 1a, overall log-rank test *p* < 0.001; pair-wise log-rank test: hepatic vs. hepatic–extrahepatic *p* = 0.001, hepatic vs. extrahepatic *p* = 0.001, extrahepatic vs. hepatic–extrahepatic *p* < 0.001).

Due to a small number at risk, two patients with extrahepatic pattern were censored after 84 months (7 years) in the Kaplan–Meier plots. There were significant differences in overall survival between AJCC stage IV categories (Figure 1b, overall log-rank test *p* < 0.001; pair-wise log-rank test: M1a vs. M1b *p* = 0.005, M1a vs. M1c *p* < 0.001, M1b vs. M1c *p* = 0.006). For each metastatic pattern, survival differences between the AJCC stage IV categories were investigated (Figure 2a-c). Significant differences in overall survival were found between the AJCC stage IV categories among patients with hepatic pattern (Figure 2a, overall log-rank test *p* < 0.001; pair-wise log-rank test: M1a vs. M1b *p* = 0.01, M1a vs. M1c *p* < 0.001, M1b vs. M1c *p* = 0.005), but not among patients with hepatic–extrahepatic pattern (Figure 2b, *p* = 0.11, overall log-rank test) or extrahepatic pattern (Figure 2c, *p* = 0.31, overall log-rank test).

No significant differences in overall survival from the date of metastatic diagnosis were found when stratifying patients by disease-free interval groups (0–3 years, 3–6 years, and >6 years) (Figure 3, *p* = 0.380, overall log-rank).

The hazard ratio (HR) for death was significantly higher in patients with hepatic pattern (HR 2.57) and hepatic–extrahepatic pattern (HR 4.36) compared to those with extrahepatic pattern in the univariate Cox regression (Table 2). The metastatic pattern remained significant in the multivariate Cox regression, with an HR of 2.37 for hepatic pattern and an HR of 3.25 for hepatic–extrahepatic pattern (Table 2).

### 3.3. Extrahepatic Lesions

The extrahepatic lesions were located in various organs (Table 3 and Table 4). Among the patients with hepatic–extrahepatic pattern, bones (45%), lungs (36%), and lymph nodes (36%) were the most common extrahepatic organs. For patients with extrahepatic pattern, lymph nodes (59%) and lungs (53%) were the most common sites, whereas bone metastases were found in 18%.

### 3.4. The Largest Diameter of the Largest Metastatic Lesion

The median LDLM was smallest in patients with hepatic pattern (27 mm [interquartile range (IQR) 18 to 50]) compared to those with hepatic–extrahepatic pattern (42 mm [IQR 30 to 73]) and extrahepatic pattern (46 mm [IQR 34–63]). The Kruskal–Wallis test showed differences between the groups (*p* = 0.005). In 93% of patients with hepatic–extrahepatic pattern, the metastasis with the LDLM was located in the liver (data available in 45 patients). In two patients with hepatic and extrahepatic pattern, the LDLM was measured in the bones (15 mm compared to 10 mm in the liver and 22 mm compared to 20 mm in the liver, respectively), and in one patient, the LDLM was measured in the spleen (34 mm compared to 30 mm in the liver).

### 3.5. Detection of Metastases: Screening vs. Symptoms

We investigated how many patients our follow-up program detected before the symptoms developed. Data were not available for all patients. A total of 61 out of 64 patients with hepatic pattern (95%) were diagnosed with metastases at a planned liver ultrasonography screening visit, whereas two patients (3%) developed symptoms that led to the diagnosis, and in one patient (2%), a metastatic liver lesion was discovered incidentally (Table 5). In 27 out of 33 patients with hepatic–extrahepatic pattern (82%), the metastatic lesions were detected at the planned liver ultrasonography screening visit, whereas six patients (18%) developed symptoms (Table 5). For the extrahepatic pattern group, 13 out of 16 patients (81%) were diagnosed with metastases due to the onset of symptoms, and for three patients (19%), uveal melanoma metastases were an incidental finding, of which one was diagnosed with metastases at the time of primary diagnosis.

### 3.6. Excluded Patients

The metastatic pattern could not be described in 71 patients, as images or imaging reports could not be retrieved (Appendix A). The median overall survival was 2.3 months in the excluded group compared to 9.5 months in the included patient group. There was a male predominance (55%), and the median age was higher (median age 69 years) in the excluded patients compared to that in the included patients (43% men, median age 66 years). A total of 60 out of the 71 excluded patients (85%) were registered with liver metastases either histopathologically or through liver ultrasonography.

## 4. Discussion

To the best of our knowledge, this study is the first to describe and investigate the impact of the anatomical location of metastatic lesions on survival, exclusively including patients who have undergone imaging assessment covering at least the chest and the abdomen. This study shows that patients with disseminated PUM without liver metastases have a significantly longer overall survival compared to patients presenting with hepatic metastases, even though a large proportion of these patients were diagnosed with metastatic disease based on symptoms and not by screening. As the Danish surveillance program includes regular liver ultrasonography, one would expect patients without liver metastases to demonstrate an even shorter survival than patients with liver metastases, as the disease will be diagnosed at a presumed later disease stage. Paradoxically, the survival was longer in the extrahepatic pattern group. A possible explanation could be that extrahepatic lesions are located at sites that are less lethal. This is supported by a higher percentage of resection of metastatic lesions among patients with hepatic–extrahepatic pattern compared to the other two groups. Another explanation could be that lesions developing exclusively in extrahepatic organs are more indolent and less aggressive than hepatic metastases. Ten percent of the patients presented with only extrahepatic metastases, which is in line with recent large cohort studies [3,4,27]. Among all the patients with hepatic metastases, patients with hepatic pattern demonstrated a longer survival than patients with hepatic–extrahepatic pattern, possibly due to a larger total tumor burden in the latter. Further, the median LDLM was smaller in patients with hepatic pattern compared to patients with hepatic–extrahepatic pattern, indicating that the latter group has more advanced disease.

It has previously been described that patients without liver involvement exhibit longer survival. Four decades ago, Bedikian et al. [17] found that 31 patients with metastatic lesions exclusively in the liver had a poorer prognosis compared to 21 patients with single-site extrahepatic metastases. Rajpal et al. [18] published a study in 1983, where they analyzed a series of 35 patients and found that the survival varied, in favor of patients with lung metastases compared to patients with liver metastases. Hepatic metastases have been associated with a poor prognosis in other previously published small cohort studies and case series [11,19,20,21,22], as well as in a more recent, larger cohort study [4]. The assessment of metastatic sites varied across the studies and did not always include whole-body imaging. In the present study, the univariate and multivariate Cox analysis both showed that the metastatic pattern was an independent prognostic factor for survival, even when adjusting for the AJCC stage, which confirms the previous findings.

In the present study, only patients who had undergone imaging of at least the chest and the abdomen were included. We used this approach to reduce the risk of missing any potential extrahepatic metastases. Consequently, due to a lack of imaging data, we had to exclude 71 out of 252 patients diagnosed with metastases. Among the excluded patients, 85% had verified liver metastases, indicating that the distribution of metastatic liver involvement is comparable between the included (90%) and excluded patients. Nonetheless, the median survival in the excluded patients was considerably shorter than in the included patients. A possible explanation could be that the excluded patients may have had a more advanced disease stage when the metastases were detected and, consequently, may have been unable or unwilling to undergo further investigation. That only 1 out of the 71 excluded patients (1%) was treated with ipi+nivo, and that there was a larger proportion of men and older patients in the extrahepatic group, supports a possible selection bias.

LDLM as a prognostic factor for survival was proposed by Eskelin et al. [16] However, it is unclear whether the LDLM applies to liver metastases only, or if it could also be applied to extrahepatic metastases. In the present study, we staged patients with extrahepatic pattern according to the AJCC 8th Edition [12], using the largest diameter of the largest metastatic lesion in the whole body (Table 1). In the total cohort, the survival varied significantly between the AJCC stages M1a, M1b, and M1c. This was also the case when only looking at patients with hepatic pattern, but there were no differences in survival between the M1a, M1b, or M1c stages within the hepatic–extrahepatic pattern or the extrahepatic pattern groups, though it should be noted that the sub-cohorts were small.

It is noteworthy that 41% of patients with hepatic metastases also had extrahepatic metastases. In patients with hepatic–extrahepatic pattern, bones were the most common extrahepatic organ (45%), which was only the case for 18% of patients with extrahepatic pattern. Lymph nodes (59%) and lungs (53%) were the most common metastatic sites in patients with extrahepatic pattern, which was also the second most commonly involved organs for hepatic–extrahepatic pattern. We were not able to identify any metastatic sub-patterns among patients with extrahepatic pattern, and larger, multicenter studies are needed to explore the frequency and distribution of metastatic sites related to survival among this small patient group. The question is whether to expand the follow-up program to include monitoring for the development of extrahepatic lesions. As the extrahepatic lesions develop in various organs, including lymph nodes and bones, surveillance for extrahepatic lesions would require whole-body imaging. We find that it would be unethical to regularly expose all patients to the radiation dose of a whole-body CT or PET/CT to find the 10% without hepatic lesions. In total, 18% of the patients with hepatic–extrahepatic pattern and 81% of the patients with extrahepatic pattern developed symptoms before being diagnosed with metastatic PUM. This emphasizes that it is important to inform all patients to be aware of symptoms from other organs than the liver and to seek medical help when experiencing any unexplained symptoms. Of the 66 patients that presented with hepatic–extrahepatic pattern, whether the first metastasis was in the liver or an extrahepatic organ is not known. However, in 93% of patients with hepatic–extrahepatic pattern, the largest metastatic lesion was found in the liver, indicating that the liver was the primary site (given that the tumor doubling time is the same in all metastatic lesions). This supports maintaining surveillance that focuses on the liver.

The disease-free interval has been suggested as a prognostic factor for survival in previous studies [13,14,16,28], where a short disease-free interval has been associated with a shorter survival from the date of metastatic diagnosis. We were not able to identify any differences in survival between patients with a short, medium, or long disease-free interval (Figure 3), and our results align with other studies that likewise have not been able to demonstrate this association [29,30]. The median disease-free survival was longer in patients with extrahepatic pattern (62.0 months, IQR 53.0–116.0) compared to those with hepatic (29.0 months, IQR 17.5–41.5) and hepatic–extrahepatic patterns (25.0 months, 12.0–66.0). As we only include patients who have developed metastases, the disease-free survival should be interpreted with caution due to selection bias. With this in mind, we still find it noteworthy that the disease-free interval was longer for the extrahepatic pattern group. A lead-time bias is likely introduced, as patients are not routinely screened for extrahepatic metastases, and patients with only extrahepatic metastases are therefore diagnosed at a later disease stage. An explanation could also be that the extrahepatic pattern represents a less aggressive tumor subtype that develops more slowly than hepatic metastases. This is highly speculative, and we do not know whether the longer disease-free interval is caused only by the lead-time bias or by a slower development and growth of extrahepatic lesions.

Ipi+nivo is the only available treatment regimen in Denmark proven to have any efficacy in patients with metastatic PUM [31,32,33]. Interestingly, in a phase II trial investigating the effect of ipi+nivo in metastatic uveal melanoma [31], 6 out of 35 patients (17%) had extrahepatic lesions only, and one of these patients (17%) experienced (as the only patient) complete remission. Three of the patients (50%) reached stable disease of at least 6 months, which was only the case for 11% of the patients with liver metastases. The authors also found a longer progression-free survival for patients with extrahepatic lesions only, though it was not significant. Another phase II study including 52 patients also investigated the efficacy of ipi+nivo and found a longer overall survival among patients with extrahepatic lesions only, though it was also not significant [32]. Whether this is caused by ipi+nivo being more effective on extrahepatic lesions or whether it is caused by extrahepatic lesions representing a less aggressive metastatic subtype is not known, but it emphasizes the need to consider the anatomical location of metastatic lesions when stratifying patients into clinical trials. It has been hypothesized that the limited effect of ipi+nivo and other immunotherapy in metastatic PUM could be caused by the high immune tolerance of the liver [34]. It has been shown that tryptophan 2,3-dioxygenase (TDO), which is predominantly expressed in the liver, is also expressed by metastatic uveal melanoma cells [34]. TDO has a suppressive effect on the immunological activity of the liver and could therefore impair the efficacy of immunotherapy.

It is still not evident why the metastases bypass the liver in some cases, but it is apparent that PUM can exhibit heterogeneous clinical courses. With genetic profiling of the primary tumor, we can quite precisely predict if the patients are at risk of developing metastases [9,35,36,37], but the genetic signatures driving the different dissemination patterns remain to be explored. The molecular profile of the primary tumor could potentially hold the key to differentiating between patients who will develop liver metastases and those who will develop metastases in other organs. This could pave the way for a more personalized follow-up program, with surveillance focusing on specific organs depending on metastatic pattern risk profiles.

## 5. Conclusions

Metastatic PUM patients with only extrahepatic lesions represented 10% of the total cohort. Despite diagnosis at the onset of symptoms, they still had significantly better overall survival compared to patients with liver metastases. Of the 90% with hepatic metastases, 41% also had extrahepatic lesions. The patients with both hepatic and extrahepatic lesions had larger liver metastases and poorer overall survival than patients with only liver metastases, indicating a more advanced stage of disease. The high percentage of liver metastases and the low incidence of symptoms among patients with only liver metastases support maintaining surveillance that focuses on the liver. The heterogenous anatomical distribution of extrahepatic lesions suggests that whole-body imaging is mandated if surveillance for extrahepatic lesions should be included; however, due to radiation exposure, we do not recommend this as a screening tool.

## Figures and Tables

**Figure 1 cancers-16-03346-f001:**
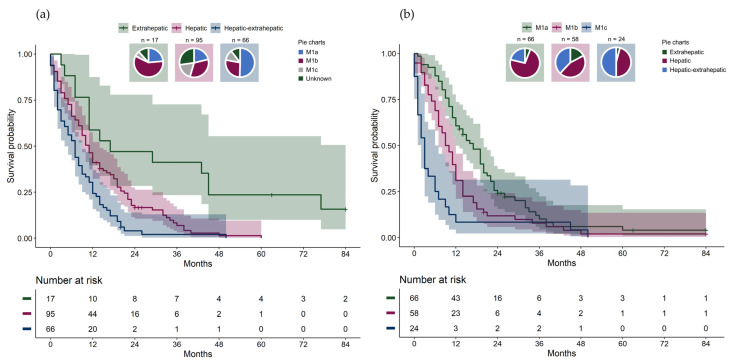
(**a**) Overall survival from metastatic diagnosis stratified by metastatic pattern with pie charts showing the distribution of American Joint Committee on Cancer (AJCC) stage IV category in each metastatic pattern group. (**b**) Overall survival from metastatic diagnosis stratified by stage IV AJCC category with pie charts showing the distribution of metastatic pattern groups in each AJCC stage.

**Figure 2 cancers-16-03346-f002:**
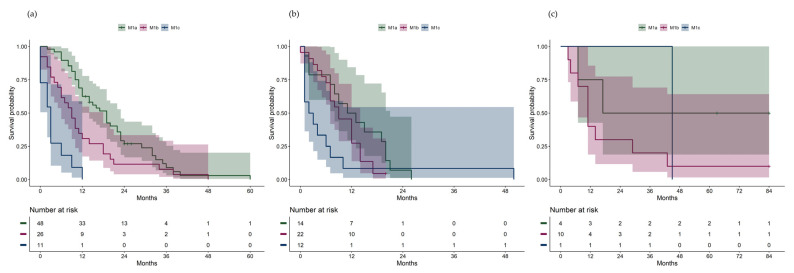
(**a**) Overall survival from metastatic diagnosis in patients with hepatic pattern, stratified by AJCC stage IV category. (**b**) Overall survival from metastatic diagnosis in patients with hepatic–extrahepatic pattern, stratified by AJCC stage IV category. (**c**) Overall survival from metastatic diagnosis in patients with extrahepatic pattern, stratified by AJCC stage IV category.

**Figure 3 cancers-16-03346-f003:**
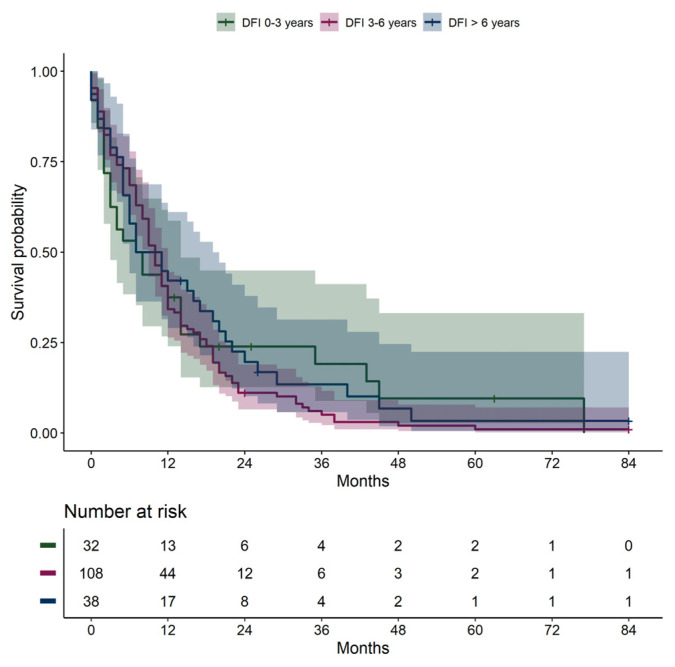
Overall survival from date of metastatic diagnosis, stratified by disease-free interval: 0–3 years, 3–6 years, >6 years. DFI, disease-free interval.

**Table 1 cancers-16-03346-t001:** Patient characteristics.

	Hepatic *n* (%)	Hepatic–Extrahepatic *n* (%)	Extrahepatic *n* (%)	Total *n* (%)
**Patients**	95 (53)	66 (37)	17 (10)	178 (100)
**Age at metastatic diagnosis (year)**				
Median (IQR)	65 (58–72)	69 (60–77)	68 (58–74)	66 (58–74)
**Age at primary diagnosis (year)**				
Median (IQR)	61 (55–70)	63 (56–72)	60 (48–69)	62 (54–71)
**Sex**				
Female	51 (54)	40 (61)	10 (59)	101 (57)
Male	44 (46)	26 (39)	7 (41)	77 (43)
**AJCC stage IV ^a^**				
M1a	48 (51)	14 (21)	4 (23)	66 (45)
M1b	26 (27)	22 (34)	10 (59)	58 (39)
M1c	11 (12)	12 (18)	1 (6)	24 (16)
Unknown	10 (10)	18 (27)	2 (12)	
**LDLM (mm) ^a^**	27 (IQR 18 to 50)	42 (IQR 30 to 73)	46 (IQR 34–63)	34 (IQR 20–60)
**ECOG performance status ^a^**				
0–1	67 (71)	38 (58)	10 (59)	115 (64)
2–4	4 (4)	10 (14)	4 (23)	18 (10)
Unknown	24 (25)	18 (27)	3 (18)	45 (25)
**First-line ipi+nivo treatment**				
No	71 (75)	54 (82)	16 (94)	141 (79)
Yes	24 (25)	12 (18)	1 (6)	37 (21)
**Resection of metastatic lesions(s)**				
No	81 (85)	65 (98)	11 (65)	157 (88)
Yes	14 (15)	1 (2)	6 (35)	21 (12)
**Median overall survival (months), (IQR)**	11.0 (5.0–20.0)	7.0 (2.0–12.0)	17.0 (11.0–45.0)	9.5 (3.0–19.0)
**Median disease-free interval (months), (IQR)**	29.0 (17.5–41.5)	25.0 (12.0–66.0)	62.0 (53.0–116.0)	29.5 (15.0–59.0)

Abbreviations: AJCC, American Joint Committee on Cancer; ECOG, Eastern Cooperative Oncology Group; ipi+nivo, first-line treatment with the combination of ipilimumab and nivolumab; IQR, interquartile range; LDLM, largest diameter of the largest metastatic lesion. ^a^ Data not available for all patients.

**Table 2 cancers-16-03346-t002:** Univariate and multivariate Cox proportional hazard model for overall survival from date of diagnosis of metastasis until death or end of follow-up.

	Univariate		Multivariate	
	HR (95% CI)	*p*	HR (95% CI)	*p*
**Sex**				
Women	1		1	
Men	1.35 (0.99–1.84)	0.057	1.51 (0.99–2.29)	0.053
**Age**				
Under 60 years of age	1		1	
Over 60 years of age	1.63 (1.16–2.30)	**0.005**	1.27 (0.78–2.06)	0.329
**Metastatic pattern**				
Extrahepatic	1		1	
Hepatic	2.57 (1.40–4.73)	**0.002**	2.37(1.08–5.17)	**0.031**
Hepatic–extrahepatic	4.41 (2.35–8.30)	**<0.001**	3.25 (1.42–7.41)	**0.005**
**AJCC stage IV ^a^**				
M1a	1		STRATA	
M1b	1.67 (1.15–2.42)	**0.006**	STRATA	
M1c	3.15 (1.93–5.14)	**<0.001**	STRATA	
**ECOG performance status**				
0–1	1		1	
>2	3.71 (2.22–6.22)	**<0.001**	2.27 (0.78–2.06)	**0.012**
**Treatment ^a^**				
No ipi+nivo	1		STRATA	
ipi+nivo treatment	0.52 (0.36–0.77)	**0.001**	STRATA	
**Resection of metastatic lesion(s)**				
No	1		1	
Yes	0.38 (0.22–0.63)	**<0.001**	0.52 (0.24–1.11)	0.092

Abbreviations: AJCC, American Joint Committee on Cancer; CI, confidence interval; ECOG, Eastern Cooperative Oncology Group; HR, hazard ratio; ipi+nivo, first-line treatment with the combination of ipilimumab and nivolumab; LDLM, largest diameter of the largest metastatic lesion. ^a^ Due to non-proportionality, the covariates AJCC stage IV and treatment with ipi+nivo were included in the multivariate Cox regression as strata.

**Table 3 cancers-16-03346-t003:** Sites and number of involved extrahepatic lesions in patients with hepatic–extrahepatic metastatic pattern.

Site	*N* (%)
Bone	30 (45)
Lungs	24 (36)
Lymph nodes	24 (36)
Skin	15 (23)
Peritoneum	11 (17)
Muscle	7 (11)
Adrenal gland	7 (11)
Pancreas	5 (8)
Spleen	4 (6)
Soft tissue	3 (5)
Thyroid	3 (5)
Breast	2 (3)
Maxillary sinus	1 (2)
Bladder	1 (2)
Ovary	1 (2)
Kidney	1 (2)
**Number of Involved Extrahepatic Organs**	
1	33 (50)
2–3	23 (35)
4 or more	10 (15)

**Table 4 cancers-16-03346-t004:** Affected extrahepatic organs in patients with extrahepatic metastatic pattern.

Patient Id	No. of Sites	Affected Organs													Symptoms	OS (M)	DFI (M)	Alive
		**Lymph nodes**	**Lungs**	**Skin**	**Soft tissue**	**Peritoneum**	**Bone**	**Brain**	**Muscle**	**Spleen**	**Gallbladder**	**Breast tissue**	**Heart atrium**	**Maxillary Sinus**				
1	6	x		x	(x)						x	x		x	Y	11	70	N
2	5	(x)	x	x	x				x						Y	11	56	N
3	4	(x)		x		x				x					Y	29	46	N
4	3	(x)	x					x							Y	3	150	N
5	3	(x)	x										x		Y	43	116	N
6	3	x	(x)			x									N	7	62	N
7	3	x	(x)			x									Y	157	58	Y
8	2	x	(x)												Y	7	47	N
9	2	x	(x)												Y	4	62	N
10	2	x					x								Y	14	112	N
11	2				x		(x)								Y	11	96	N
12	1		(x)												N	59	188	Y
13	1						(x)								N	116	0	Y
14	1			(x)											Y	17	40	N
15	1							(x)							Y	77	135	N
16	1		(x)												Y	45	132	N
17	1									(x)					NA	45	53	N
Organs, total		10	8	4	3	3	3	2	2	2	1	1	1	1				

Affected organs (marked with x) at the time of metastatic diagnosis in the 17 patients with exclusive extrahepatic metastases. (x) indicates where the metastatic lesion with the largest diameter is located (not available in patient number 10). The metastatic diagnosis was verified histopathologically in all patients with extrahepatic pattern, but not all affected organs were biopsied. Abbreviations: DFI, disease-free interval; M, months; N, no; NA, not available; OS, overall survival; Y, yes.

**Table 5 cancers-16-03346-t005:** The cause leading to the detection of the first metastatic lesion, either at a scheduled liver ultrasonography visit, due to symptoms, or as an incidental finding while investigating for other diseases.

Detection of Metastases			
	**Liver Ultrasonography Surveillance**	**Symptoms**	**Incidental Finding**
Hepatic metastases (*n* = 64)	61 (95%)	2 (3%)	1 (2%)
Hepatic–extrahepatic metastases (*n* = 33)	27 (82%)	6 (18%)	0 (0%)
Extrahepatic metastases (*n* = 16)	-	13 (81%)	3 (19%)

Data are not available for all patients.

## Data Availability

Data cannot be fully anonymized and therefore cannot be made publicly available as per the European General Data Protection Regulation (GDPR). Inquiries regarding data can be directed to J.F.K. (jens.folke.kiilgaard@regionh.dk).

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
