# Peer review of "Impact of Metastatic Pattern on Survival in Patients with Posterior Uveal Melanoma: A Retrospective Cohort Study"

_cancers, 2024, doi:10.3390/cancers16193346_

Round 1

Reviewer 1 Report

Comments and Suggestions for Authors

Authors conducted research on patients with Metastatic posterior uveal melanoma (PUM) and investigated if the anatomical location of the metastatic lesions was associated with differences in survival. 178 patients with metastatic PUM were included. 53% presented with hepatic pattern, 37% presented with hepatic-extrahepatic pattern, and 10%  with extrahepatic pattern. Overall survival was significantly longer in patients with extrahepatic pattern (median 17.0 months), compared to hepatic pattern and hepatic-extrahepatic pattern.

Minor comments.

Introduction: Lines 51-52. Authors start with survival time of patients with metastatic posterior (choroidal and ciliary body) uveal melanoma (PUM). However, PUM is rare disease and many readers are not familiar with it.  I recommend to start the introduction giving more information on this cancer type.

Furthermore, authors should mention how often PUM is metastatic. This melanoma is in the eye- are the main metastases outside of the head?

What are risk factors for this kind of melanoma and what is the prevalence?

Abstract: I recommend to include Hazard Ratios and 95% CI in results.

Author Response

The authors would like to thank the reviewer for the constructive feedback. Please find our responses below.

Comment 1: "Introduction: Lines 51-52. Authors start with survival time of patients with metastatic posterior (choroidal and ciliary body) uveal melanoma (PUM). However, PUM is rare disease and many readers are not familiar with it.  I recommend to start the introduction giving more information on this cancer type."

Response 1: Thank you for pointing this out. We agree with this comment and have included a section with introduction of posterior uveal melanoma to the Background section (lines 53-61). Additional references are added to this section. For further elaboration, please see the following comments.

Comment 2: "Furthermore, authors should mention how often PUM is metastatic. 

Response 2: We agree with this comment. We have added a paragraph to the Background section stating how often PUM metastasizes: "... about half of the patients die due to metastatic disease [5]" (lines 57-58).

Comment 3:This melanoma is in the eye- are the main metastases outside of the head?"

Response 3: Thank you for this comment. Yes, the vast majority of patients develop liver metastases, and brain metastases are very rare. We have added a paragraph to the Background section (lines 56-57) stating : "PUM has a high tendency to metastasize, primarily to the liver [3,4]"

The high occurrence of hepatic metastases is also emphasized in the Results section (lines 148-151): "Ninety-five patients (53%) had metastatic disease exclusively to the liver (hepatic pattern), 66 patients (37%) had both hepatic and extrahepatic metastases (hepatic-extrahepatic pattern), and 17 patients (10%) had exclusive extrahepatic metastases (extrahepatic pattern)" and in Table 1. Information about brain metastases can be found in Table 3 and 4. Table 3 shows affected extrahepatic organs in patients with both hepatic and extrahepatic metastases and Table 4 shows affected organs in patients with exclusively extrahepatic metastases. Only two patients with extrahepatic pattern (Table 4) have brain metastases. 

Comment 4: What are risk factors for this kind of melanoma and what is the prevalence?

Response 4: We agree with this comment and have added a paragraph about the risk factors for metastatic PUM (lines 58-60): The size of the eye tumor, extraocular growth, ciliary body involvement, and chromosomal and genetic alterations in the primary tumor, such as monosomy 3 and mutation in BAP1, increase the risk of metastasis [6–9].

As uveal melanoma is a rare disease, the frequency is generally reported as incidence rates. We have added a paragraph about reported incidence rates (lines 54-56): It is the most common primary intraocular malignancy, with the highest incidence rates above 8.0 cases per million person-years occurring in Northern European countries and in Australia and New Zealand [1,2]. 

Comment 5: Abstract: I recommend to include Hazard Ratios and 95% CI in results.

Response 5: We agree with this comment. We have added a paragraph reporting the multivariate hazard ratios with 95% CI in the abstract (lines 41-43): Multivariate Cox regression analysis showed increased hazard ratios (HR) for hepatic pattern (HR 2.37, 95% CI 1.08-5.17, p = .031) and hepatic-extrahepatic pattern (3.25, 95% CI 1.42-7.41, p = .005)., compared to extrahepatic pattern.

Reviewer 2 Report

Comments and Suggestions for Authors

Overall very interesting findings by the authors. It is very striking to note that the majority of the metastases occurs to the liver in the cases studied. The authors must discuss the immensely important role played by TDO in metastatic uveal melanoma (PMID: 32050636). TDO is a heme protein whose heme insertion mechanisms were recently discovered (PMID: 35051612) and nitric oxide was found to regulate the heme levels and hence the activity of TDO (PMID: 37116709). Intriguingly, TDO is expressed in the liver to breakdown L-Trp to Kynurenine (PMID: 14794727) and is involved in immune suppression in various cancer cells through formation of L-Kyn which binds to AhR and suppresses the differentiation of T effector cells and promotes the differentiation of T reg cells (PMID: 27773992). Another possible reason for the limited success of the Ipi+nivo therapy is due to the presence of the Kyn pathway which is controlled by TDO/IDO1. Although nivo binds to PD1 on the T cell surface so that it cannot bind to PDL1 on the cancer cells to trigger immune suppression, the cancer cells are also expressing TDO/IDO1 which will breakdown L-Trp and generate Kynurenine which is secreted out of the cancer cells, enter the T cells, binds to AhR and inhibit T effector cell differentiation. This back-up mechanism involving TDO/IDO1 and L-Trp breakdown would greatly limit the efficacy of Ipi+nivo therapy as the authors are also observing.

The authors must discuss the above mentioned points in their discussion section and include the supporting literature which would greatly enrich the manuscript and also help explain their findings. 

Author Response

The authors would like to thank the reviewer for the constructive feedback. Please find our responses below.

Comment 1: Overall very interesting findings by the authors. It is very striking to note that the majority of the metastases occurs to the liver in the cases studied. The authors must discuss the immensely important role played by TDO in metastatic uveal melanoma (PMID: 32050636). TDO is a heme protein whose heme insertion mechanisms were recently discovered (PMID: 35051612) and nitric oxide was found to regulate the heme levels and hence the activity of TDO (PMID: 37116709). Intriguingly, TDO is expressed in the liver to breakdown L-Trp to Kynurenine (PMID: 14794727) and is involved in immune suppression in various cancer cells through formation of L-Kyn which binds to AhR and suppresses the differentiation of T effector cells and promotes the differentiation of T reg cells (PMID: 27773992). Another possible reason for the limited success of the Ipi+nivo therapy is due to the presence of the Kyn pathway which is controlled by TDO/IDO1. Although nivo binds to PD1 on the T cell surface so that it cannot bind to PDL1 on the cancer cells to trigger immune suppression, the cancer cells are also expressing TDO/IDO1 which will breakdown L-Trp and generate Kynurenine which is secreted out of the cancer cells, enter the T cells, binds to AhR and inhibit T effector cell differentiation. This back-up mechanism involving TDO/IDO1 and L-Trp breakdown would greatly limit the efficacy of Ipi+nivo therapy as the authors are also observing.

The authors must discuss the above mentioned points in their discussion section and include the supporting literature which would greatly enrich the manuscript and also help explain their findings. 

Response 1: 

Thank you to the reviewer for suggesting a relevant topic to enrich our Discussion section. The overall aim of this study was to investigate clinical metastatic patterns in metastatic posterior uveal melanoma and their relation to survival. The aim of the study was not to explore the biological mechanisms on a cellular level that could explain this variability in dissemination patterns. However, we acknowledge that it is highly interesting that TDO, expressed by liver cells and metastatic uveal melanoma cell, has a suppresive effect on the immunological activity, which could explain why immune check point inhibitors have limited effect in metastatic uveal melanoma. In the Discussion section, we discuss the efficacy of ipi+nivo in metastatic uveal melanoma and highlight two randomized studies on ipi-nivo treatment in metastatic posterior uveal melanoma, in which the authors find better outcomes in patients with sole extrahepatic lesions. To this section, we have added a few paragraphs about TDO and the expression in normal liver and metastatic posterior uveal melanoma cells in the liver and refer to the aforementioned article by the reviewer: (PMID: 32050636: Terai, M.; Londin, E.; Rochani, A.; Link, E.; Lam, B.; Kaushal, G.; Bhushan, A.; Orloff, M.; Sato, T. Expression of Tryptophan 2,3-Dioxygenase in Metastatic Uveal Melanoma. Cancers (Basel). 2020, 12, 0–14, doi:10.3390/cancers12020405.):

Lines 376-381 have been added: "It has been hypothesized that the limited effect of ipi+nivo and other immunotherapy in metastatic PUM could be caused by the high immune tolerance of the liver [34]. It has been shown that tryptophan 2,3-dioxygenase (TDO), which is predominantly expressed in the liver, also is expressed by metastatic uveal melanoma cells [34]. TDO has a suppressive effect on the immunological activity of the liver and could thereby impair the efficacy of immunotherapy."

We agree with the reviewer that this substantiates the points made in the discussion. We find that the other suggested articles address the topic at a very in-depth cellular level, which we consider to be out of scope for this article.